# Epithelial Ovarian Cancer: Providing Evidence of Predisposition Genes

**DOI:** 10.3390/ijerph19138113

**Published:** 2022-07-01

**Authors:** Sidrah Shah, Alison Cheung, Mikolaj Kutka, Matin Sheriff, Stergios Boussios

**Affiliations:** 1Department of Palliative Care, Guy’s and St Thomas’ Hospital, London SE1 9RT, UK; sidrah.shah@nhs.net; 2Department of Hematology/Medical Oncology, Medway NHS Foundation Trust, Windmill Road, Kent, Gillingham ME7 5NY, UK; alison.cheung2@nhs.net (A.C.); mikolaj.kutka@nhs.net (M.K.); 3Department of Urology, Medway NHS Foundation Trust, Windmill Road, Kent, Gillingham ME7 5NY, UK; matin.sheriff@nhs.net; 4King’s College London, Faculty of Life Sciences & Medicine, School of Cancer & Pharmaceutical Sciences, London SE1 9RT, UK; 5AELIA Organization, 9th Km Thessaloniki-Thermi, 57001 Thessaloniki, Greece

**Keywords:** ovarian cancer, hereditary, *BRCA1/2* genes, poly(ADP-ribose) polymerase (PARP) inhibitors, next-generation sequencing (NGS)

## Abstract

Epithelial ovarian cancer (EOC) is one of the cancers most influenced by hereditary factors. A fourth to a fifth of unselected EOC patients carry pathogenic variants (PVs) in a number of genes, the majority of which encode for proteins involved in DNA mismatch repair (MMR) pathways. PVs in *BRCA1* and *BRCA2* genes are responsible for a substantial fraction of hereditary EOC. In addition, PV genes involved in the MMR pathway account for 10–15% of hereditary EOC. The identification of women with homologous recombination (HR)-deficient EOCs has significant clinical implications, concerning chemotherapy regimen planning and development as well as the use of targeted therapies such as poly(ADP-ribose) polymerase (PARP) inhibitors. With several genes involved, the complexity of genetic testing increases. In this context, next-generation sequencing (NGS) allows testing for multiple genes simultaneously with a rapid turnaround time. In this review, we discuss the EOC risk assessment in the era of NGS.

## 1. Introduction

Ovarian cancer is the fifth leading cause of cancer-related mortality in females [1]. Over two-thirds of women present with advanced stages of ovarian cancer, which leads to an estimated 5-year survival rate of between 20–40% [2]. In contrast, those diagnosed at an earlier stage (e.g., stage 1 disease) have a 5-year survival rate of over 90% [3]. The GLOBOCAN study predicted a worldwide increase of 55% in ovarian cancer cases and a 67% increase in deaths from the year 2012 to 2035 [4]. The median age of women at diagnosis is 63 years [5]. Epithelial ovarian cancer (EOC) is a heterogenous disease further classified into benign, borderline, and malignant [6,7]. Malignant EOC includes five main histological subtypes: high-grade serous ovarian cancer (70–80%), endometrioid (10%), clear cell (10%), mucinous (3%), and low-grade serous (<5%) [6,7]. Each subtype behaves as a distinct disease with differences in clinical presentations, mutations, and responses to treatment such as chemotherapy [8]. It is well-established that EOC develops according to two different carcinogenic pathways. The vast majority of these tumours are high-grade serous tumours that develop according to the type II pathway and present *p53* and *BRCA* mutations. In contrast, low-grade serous tumours are characterised by *KRAS*, *BRAF*, *PTEN*, *PIK3CA*, *CTNNB1*, *ARID1A*, and *PPP2R1A* mutations and progress according to the type I pathway [9]. Beyond EOC, there are also non-epithelial counterparts that are further divided into germ cell (5%) and sex-cord stromal cell (5%) ovarian cancers [10,11,12]. Ovarian carcinosarcoma is even rarer; it is a biphasic, but challenging histologic subtype, accounting for only 1–4% of all ovarian cancers [13].

Newly diagnosed high-grade serous EOC patients are treated with radical surgery followed by adjuvant platinum and taxane combination chemotherapy. In EOC patients where upfront surgery is contraindicated for medical reasons, or where complete cytoreduction cannot be achieved, neoadjuvant chemotherapy prior to interval debulking surgery followed by adjuvant chemotherapy is an alternative therapeutic option [14]. The treatment of gestational ovarian malignancies depends on the histology, stage, and gestational weeks. When possible, surgical excision is indicated; fertility-sparing surgery can be offered to stage I EOC, germ cell ovarian, or sex-cord stromal ovarian tumours [15].

Approximately 20–30% of EOC occurs in females with an inherited predisposition; most of these hereditary ovarian cancers are due to germline mutations in *BRCA1* and *BRCA2* genes [16,17,18,19]. Lynch/hereditary nonpolyposis colorectal cancer (HNPCC) syndrome is also associated with an increased risk of ovarian cancer and is caused by germline mutations in the mismatch repair (MMR) genes [20,21]. The *BRCA1* and *BRCA2* genes are involved in the repair of double-strand DNA breaks (DSBs) through the homologous recombination (HR) system [22]. Pathogenic variants (PVs) in these genes cause a dysfunction in the BRCA proteins and the HR system, and increase the risk of ovarian cancer [22,23]. Carriers of germline mutations in *BRCA1* and *BRCA2* genes have a lifetime risk of ovarian cancer of 35–60% and 12–25%, respectively, along with an increased risk of peritoneal and fallopian tube malignancies [24,25,26]. Therefore, the identification of these germline *BRCA1/2* mutations has implications for the therapeutic management of EOC [16]. *BRCA1/2* carriers respond well to platinum-based chemotherapy and poly(ADP-ribose) polymerase (PARP) inhibitors [27,28,29]. New genes (*RAD51C*, *RAD51D*, and *PALB2*) have been identified as increasing susceptibility to ovarian cancer, but further research is required to investigate this [30,31,32,33]. In this review, we discuss susceptibility to EOC, testing, and EOC risk assessments in the context of next-generation sequencing (NGS).

## 2. Molecular Landscape

DNA damage occurs daily in cells and there are complex repair pathways to prevent the genomic instability that can be predisposed to cancer [34,35,36]. HR is a repair pathway for complex DNA damage, including DSBs and DNA gaps [37]. *BRCA1* and *BRCA2* genes are integral components of the HR repair pathway, acting as tumour suppressors and preserving the chromosome structure [23,38]. As part of the HR pathway, the BRCA2 protein recruits the recombinase RAD51 to the DSBs by forming a BRCA1–PALB2–BRCA2 complex [39]. Germline mutations in both *BRCA1* and *BRCA2* can, therefore, lead to a deficiency in the HR pathway and increase susceptibility to ovarian cancer [38]. Identifying these mutations is significant as those with *BRCA1/2* mutations are sensitive to a therapy with PARP inhibitors [40,41,42,43]. Combinations of PARP inhibitors with drugs that inhibit HR may sensitise EOC with a primary or secondary HR proficiency to PARP inhibitors and potentially expand their use beyond HR-deficient ovarian cancers. Regarding this, PARP inhibitors may be combined separately with anti-angiogenics and immune checkpoint inhibitors as well as with phosphoinositide 3-kinase (PI3K), protein kinase B (AKT), mammalian target of rapamycin (mTOR), WEE1, mitogen-activated protein kinase (MEK), and cyclin dependent kinase (CDK) 4/6 inhibitors, or even with standard chemotherapy [44].

If the HR repair pathway is impaired, synthetic lethality may be induced by a PARP inhibition exclusively for the target tumour tissue and spare normal cells [2,29,45]. The PARPs are a group of 18 multifunctional enzymes, the most active of which are PARP1 and PARP2; these repair DNA single-strand breaks through a base excision repair [29,46,47]. PARP inhibition leads to accumulating single-strand DNA breaks (SSBs) and subsequently leads to DSBs at DNA replication forks [29]. These breaks are usually repaired by a functioning HR pathway [48]. Cells carrying the heterozygous *BRCA1/2* mutation result in tumours that carry a DNA mismatch repair (MMR) deficiency unlike normal cells [29]. Therefore, PARP inhibition can selectively target the tumour tissue and lead to unrepaired SSBs, which then leads to an accumulation of DSBs and collapsed replication forks as well as chromosomal instability and consequent tumour cell apoptosis [29,49,50,51]. This synthetic lethality—a phenomenon where multiple, concurrent genetic events lead to cell apoptosis rather than a single event—allows the increased sensitivity of *BRCA*-mutated tumours to PARP inhibitors and is a novel approach that has changed the treatment of ovarian cancer [52,53]. Figure 1 provides a schematic of the roles of *BRCA1* and *BRCA2* in the DNA repair mechanism.

Apart from breast and ovarian cancer, several PARP inhibitors have been investigated in metastatic castration-resistant prostate cancer patients [54]. They are particularly effective in *BRCA1/2* mutations with increased survival outcomes. Olaparib is used in this subset of patients after a progression on novel hormonal agents, e.g., enzalutamide or abiraterone, whilst rucaparib is considered in combination with androgen receptor-guided therapy and paclitaxel-based chemotherapy [55]. Moreover, PARP inhibitor monotherapy induces an objective anti-tumour activity in patients with *PALB2*, *BRIP1*, or *FANCA* aberrations. In contrast, those with *ATM* and *CDK12* alterations do not seem to benefit [56].

MMR genes are another class of genes that are linked to a susceptibility to ovarian cancer [3]. The MMR system involves seven genes: *MLH1*, *MSH2*, *MSH6*, *PMS2*, *PMS1*, *MSH3* and *MLH3*; these act sequentially to identify MMR mutations and then form a protein complex to correct them [57]. *MSH2* and either *MSH3* or *MSH6* form a complex to identify the mismatch; this then binds to a complex of *MLH1* with either *PMS2*, *PMS3*, or *MLH3* to repair the defect [57]. An impaired MMR function through MMR mutations leads to microsatellite instability (MSI); this is characteristic of HNPCC, the third most common hereditary ovarian cancer [57]. This autosomal dominant condition is most often associated with germline mutations in the *MSH6*, *MSH2*, and *MLH1* genes and increases the risk of the endometrioid and clear cell subtypes of ovarian cancer [6,57,58,59]. The cumulative lifetime risk of ovarian cancer in women with HNPCC is estimated at 4 to 12% and the prognosis is linked to the MMR variants [6,57]. Furthermore, the mean age of diagnosis with these germline MMR mutations is 9 to 13 years earlier than the general population with sporadic tumours [60]. Increasing research to improve the knowledge of the genetic mechanisms associated with hereditary ovarian cancer is, therefore, paramount for the future of its management.

There are five major histologies of EOC: high-grade serous, low-grade serous, mucinous, endometrioid, and clear cell [61]. Mutations in the DNA repair genes differ across different histological subtypes. For example, mutations in *BRCA1/2* genes are often found in high-grade serous, but rarely in mucinous, endometrioid, and clear cell EOC [62,63,64]. The Cancer Genome Atlas project identified that 16% of high-grade serous EOC patients had germline *BRCA1/2* mutations [65]. Sugino et al., investigated these differences further in high-grade serous, endometrioid, and clear cell EOC; a few of the common mutations are summarised in Table 1 [62]. Furthermore, MMR mutations were also investigated in these histologies. MMR mutations were found in 10% of endometrioid, 3% in clear cell, and 2% of high-grade serous EOC [62]. Lheureux et al., also found that MMR mutations accounted for 10–15% of hereditary ovarian cancer, especially in endometrioid and clear cell histological subtypes [66]. Norquist et al., found that mutations were similar in high-grade serous, endometrioid, and clear cell EOC, but there was an overall lower mutation frequency in low-grade compared with high-grade serous EOC (5.7% vs. 19.6%, respectively; *p* = 0.003) and there were no ovarian cancer-associated mutations in the mucinous histology [18].

## 3. Susceptibility to EOC

The parts of the gene responsible for an increased susceptibility or predisposition to cancer in individuals are called PVs [67]. In the case of ovarian cancer, at least one of these variations in the actionable genes was found in 49.2% of patients [67]. Moreover, the prevalence of PVs is said to increase with the number of primary cancers (PCs): 13.1% with 2 PCs, 15.9% with 3 PCs, and 18.0% with ≥4 PCs [68]. This shows that EOC has a higher incidence compared with the general subset of malignant malformations. There are plenty of well-established PVs in ovarian cancer that have been shown to have a significant correlation with EOC such as *BRCA1* and *BRCA2* [69]. There are studies that show promising results for other PVs that are connected to EOC and have shown a potential for diagnostic and treatment use. Several of them include MMR genes; e.g., *MSH6*, *MSH2*, and *MLH1* [70,71].

### 3.1. Germline Predisposing Variants

#### 3.1.1. *BRCA1* and *BRCA2* Genes

The highly penetrant genes *BRCA1* and *BRCA2* are associated with most of the identified germline mutations in EOC patients. These genes produce proteins involved in fundamental cellular processes such as cell cycle checkpoint control, chromatin remodelling, transcriptional regulation, and mitosis [72]. Therefore, an HR deficiency caused by these mutations is often utilised in the treatment of EOC with platinum-based chemotherapy or PARP inhibitors [73]. Apart from EOC, these germline variants have been shown to be associated with 22 first PCs [69]. However, these germline mutations are the strongest-known genetic risk factors for EOC and are found in 6–15% of women with EOC [74]. The *BRCA1/2* status can be used by healthcare professionals for patient counselling regarding expected survival as *BRCA1* and *BRCA2* carriers with EOC respond better than non-carriers to platinum-based chemotherapies. This yields greater survival, even though the disease is generally diagnosed at a later stage and higher grade [75]. Figure 2 provides the distribution of germline PVs identified in unselected EOC patients. A fourth to a fifth of these patients carried PVs in a number of genes, the majority of which encode for proteins involved in the DNA repair pathways.

#### 3.1.2. RAD51C, RAD51D, and BRIP1 Genes

Collectively, germline mutations in *BRIP1*, *RAD51C*, and *RAD51D* account for around 2% of ovarian cancer cases [76]. Despite significant evidence of a strong correlation with EOC, the risk attributed to particular genes varies substantially among studies (odds ratio (OR) values estimated for *BRIP1* ~ 5–19, *RAD51C* ~ 5–15, and *RAD51D* ~ 6–12) [77,78,79]. BRCA1-interacting protein C-terminal helicase 1 (*BRIP1*), RAD51 homolog C (*RAD51C*), and RAD51 paralog D (*RAD51D*) are all coding for proteins that interact with BRCA1/2 and support the MMR process. Hence, patients with germline mutations in *BRIP1*, *RAD51C*, and *RAD51D* would also benefit from a therapy with PARP inhibitors.

*RAD51C* and *RAD51D* are essential for HR repair [30,80]. Mutations in these genes have a higher likelihood of high-grade serous EOC in women aged between 40 and 49 years old [60]. These genes have a few similarities with the *BRCA1/2* genes, but there is little evidence to suggest they also increase the risk of breast cancer as with *BRCA1/2* [60]. Furthermore, the location of their mutation is not variable as with *BRCA1/2*, which determines the risk of ovarian cancer [81]. Mutations in *RAD51C* occur between amino acid 143 and 319, which affect the RAD51B–RAD51C–RAD51D–XRCC2 and RAD51C–XRCC3 complexes; mutations in *RAD51D* occur in the C-terminal region, which affect binding to *RAD51C* and DSB repair [82].

Mutations in the *BRIP1* gene have been found in the first two-thirds of the gene between nucleotides 68 and 2508; these cut the protein before the BRCA1 binding domain [77]. This domain is found in proteins acting as checkpoints for DNA damage; losing this interaction impairs the DNA damage repair in the epithelial cells in the ovaries or fallopian tubes [83,84,85]. Furthermore, an increased risk of BRIP1 missense variants has been associated with high-grade serous EOC [86]. Several countries have implemented recommendations such as those for *BRCA1/2* carriers where *BRIP1*, *RAD51C*, and *RAD51D* mutation carriers are given an option of a salpingo-oophorectomy beginning at age 45–50 as a prophylactic measure [77].

#### 3.1.3. PALB2 Gene

Although there are studies that identify *PLAB2* as one of the PVs in EOC, a meta-analysis showed a non-significant increased risk of ovarian cancer (OR = 4.55; 95% confidence interval (CI), 0.76–27.24; *p* = 0.10) [71]. Moreover, it showed that germline *PALB2* mutations were rare; *PALB2* mutation carriers were found in only 0.21% of the ovarian cancer patients analysed, which is not significantly different from the frequency in the general population (0.05%) [84]. More research needs to be undertaken to further investigate the connection.

#### 3.1.4. MLH1, MSH2, MSH6, and PMS2 Genes

DNA MMR consists of three stages: initiation, excision, and resynthesis. MutS homologs (*MSH2*, *MSH3*, and *MSH6*) are a few of the proteins involved in the initiation stage of MMR [87]. Lynch syndrome, also called HNPCC, is caused by mutations in these variants (*MLH1*, *MSH2*, *MSH6*, and *PMS2*). EOC patients with Lynch syndrome are often younger with a median age of diagnosis of 43 years; however, the prevalence of the disease in this syndrome is small at 0.9–2.7% [88,89]. Pal et al., reported that clear pathogenic germline *MLH1*, *MSH2*, and *MSH6* mutations occurred in only 9/1893 (0.5%) of unselected EOC patients [90]. A small number of studies have investigated the survival of women with EOC due to MMR defects and the results are inconclusive [91]. This contrasts with the research undertaken in colorectal cancer, where MMR mutations are wildly established.

## 4. Tumour Testing in EOC

### 4.1. NGS for Cancer Risk Assessments

NGS technology has rapidly evolved over the last few decades. This revolutionary sequencing technique tests large genomic regions with a single test in a short period of time without needing a prior knowledge of the gene sequence [92]. NGS has almost replaced Sanger’s sequencing, which can only sequence a particular genomic region of interest due to the large workload and high costs [67]. Nowadays, multigene panels that analyse selected genes of interest are the most frequently used sequencing method in clinical applications [93]. Commonly, the included genes for testing are highly penetrant; however, they can also include moderate- and low-penetrance PV genes, depending on the actionability and significance of the pathological variant [94,95]. It provides valuable clinical information such as susceptibility to cancers and hereditary cancer syndromes in a cost-effective manner.

It also limits the risk of gathering large amounts of variants of unknown significance (VUS) that do not have enough data to change the clinical management. Gene panels with NGS also have a high accuracy rate in detecting all classes of mutations and genetic alterations in patients with suspected cancers [33,96]. This has significant clinical implications for patients with known hereditary gynaecological cancers who have a negative single-gene test as gene panels can often identify pathogenic mutations outside the high-penetrant genes [97]. Multigene panel testing has a much faster turnaround time and is much more cost-effective compared with single-gene testing [98,99]. It also reduces the chance of missing a mosaicism of pathogenic gene mutations in patients with hereditary cancer as NGS can identify even low levels of mutations [16].

### 4.2. Upfront Tumour Testing

Currently, tumour sample testing for ovarian cancer involves testing for *BRCA1/2* mutations and HR deficiency status, which predicts the response to platinum agents and PARP inhibitors [38]. If a mutation is detected, sequencing is performed on normal cells to determine if the mutation is somatic or germline. Considering the above, performing tumour sequencing upfront with NGS can possibly avoid additional testing as it provides information on both the HR deficiency status and the gene mutation profile (germline or somatic). It also rules out mutations in susceptibility genes [100]. Recent studies have demonstrated that tumour sequencing has a high accuracy [101,102,103]. A tumour-first testing approach can facilitate early treatment decision-making for the use of PARP inhibitors and early cancer prevention, given that more than 20% of EOC patients have a hereditary link [33].

### 4.3. T Cell Repertoire Sequencing

There is an immune response to tumours that is facilitated by tumour-infiltrating T lymphocytes (TILs) [104]. These TILs are regulatory T cells that respond to shared tumour antigens [105]. Deep sequencing of T cell receptor genes has been used to identify TILs in ovarian cancer to compare the T cells in the blood. TIL repertoires within ovarian tumours are discrete from the T cell repertoire circulating in the peripheral blood. This makes TILs a prognostic marker and their presence (e.g., CD3^+^ and CD8^+^) has been shown to improve overall survival [104]. CD8^+^ TILs have particularly shown an increased survival benefit than CD3^+^ TILs. Hwang et al., suggest that CD8^+^ staining should be used to measure TIL proportions in ovarian cancer [106]. Zhang et al., found that ovarian tumours containing these T cells had a progression-free survival of 22.4 months and an overall survival of 50.3 months compared with 5.8 and 18 months, respectively, for tumours with no TILs [107]. Better outcomes with TILs indicate that they have an anti-tumour effect and reduce growth through the release of specific cytokines, e.g., interferon-gamma, which promotes inflammation and tumour elimination [107,108]. The Elispot (enzyme-linked immunospot) assay can detect antigen-specific T cells and this can detect TILs in the peripheral blood of patients [107]. This opens possibilities for testing using T cell repertoires from TILs. It also has implications for treatments with immunotherapy. There have been mixed results with the use of TILs in ovarian cancer. In advanced or recurrent ovarian cancer, Aoki et al., found clinical responses in five out of seven patients receiving TILs alone and in nine out of ten patients receiving chemotherapy only [109]. In other studies, there were no significant clinical responses and using TILs did not enhance the immune response [110,111]. Further studies need to be completed to conclude whether TIL therapy can be implemented alongside chemotherapy with the previously observed clinical significance practically translated into better survival outcomes.

### 4.4. Pre- and Post-Test Counselling

Genetic counselling is known to help patients decrease their anxiety and depression about the cancer diagnosis and the frequency of testing [112]. In the era of NGS, genetic counselling models must evolve alongside the rapidly changing genetic testing technologies and ovarian cancer management strategies. Despite current guidelines recommending that all women with a diagnosis of EOC should undergo genetic testing at the time of the diagnosis, only approximately 10–30% of these women are referred for germline genetic testing [113,114,115,116]. To ensure genetic counselling is delivered in a timely manner to avoid missed therapeutic and prevention opportunities, the authors of a Canadian review article on genetic assessments for BRCA-associated malignancies recommended a tumour-first testing model to detect both somatic and germline *BRCA1/2* cases and to ensure that this is available to all patients without depending on a referral system [117].

With a multigene panel, the rates of VUS increase along with the detection rate [38]. A large retrospective study assessed individuals who underwent genetic testing over a decade and found that 91% of VUS were reclassified as benign or likely benign and their penetrance and cancer risks remained unknown [118]. The lack of actionability of VUS is often very distressing to patients [69]. Hence, in pre-test counselling, patients should be fully informed of the genes that will be tested, the level of penetrance, and the implications in cancer predisposition and clinical management. Fecteau et al., recommended that tested PV genes should be categorised into high, moderate, and unknown penetrance to help patients to understand the cancer risk profile of the genetic mutations, the understanding of them of scientists, and their clinical significance [119].

Post-test counselling should be guided by the findings of the test. The implications of the mutations should be explained to patients as well as their family members. Cascade testing for relatives of mutation carriers should also be recommended, if applicable [16]. It is essential that genetic counsellors fully explain what VUS results mean and their limitations. Patients should understand that although scientists currently have a limited knowledge of VUS, our understanding of them will improve with more data available in time and with international collaboration; hence the importance of regular follow-ups in the future.

### 4.5. Treatment and Recurrent Ovarian Cancer

As mentioned previously, different histologies of EOC may have different mutations; however, they have traditionally been grouped into one unit [120,121]. Most of the hereditary ovarian cancers are linked to BRCA1/2 mutations, especially in high-grade serous EOC. An analysis allows for genetic testing in relatives; if identified in asymptomatic carriers, preventative measures can be offered to reduce their risk of ovarian cancer [122]. Otherwise, if negative, they can be reassured their risk is similar to the general population risk. Risk-reducing options in BRCA1/2 carriers include a bilateral salpingo-oophorectomy (BSO), which is invasive compared with regular surveillance; that is, it is not always reliable in identifying the early stages [122]. A BSO is performed between the ages of 35 and 40 in BRCA1 carriers and 40 and 45 in BRCA2 carriers due to the late onset and can reduce the risk of ovarian cancer by 96% [123,124,125].

Careful counselling is required with planned pregnancies; menopausal symptoms can be induced, which have their own implications on quality of life and physical health (e.g., cardiovascular), and the use of hormone replacement therapy [123]. An analysis of mutations also allows tailored treatment planning, including PARP inhibitors, which have reformed ovarian cancer management [122]. Studies have found improved progression-free survival with olaparib and an efficacy of niraparib and rucaparib as maintenance treatments [122,126]. These studies examined recurrent EOC in particular and are summarised in Table 2.

**Table 2 ijerph-19-08113-t002:** Studies of PARP inhibitors in recurrent EOC.

Study/Reference	Population	Treatment Plan	Median PFS
SOLO 1/[127]	Stage III or IV high-grade serous or endometroid cancer with *BRCA1/2* mutation and complete/partial response to CTH	300 mg olaparib b.d. or placebo	49.9 vs. 13.8 m (*p* < 0.0001)
SOLO 2/[126]	Relapsed, platinum-sensitive EOC with a *BRCA1/2* mutation	300 mg olaparib b.d. or placebo	19.9 vs. 5.5 m (*p* < 0.0001)
SOLO 3/[128]	Germline *BRCA*-mutated ovarian cancer with relapse ≤ 12 m	300 mg olaparib b.d. or non-platinum CTH of the choice of the physician	13.4 vs. 9.2 m (*p* = 0.013)
NOVA/[129]	Platinum-sensitive recurrent EOC	Niraparib 300 mg o.d. or placebo	Germline *BRCA* cohort: 21.0 vs. 5.5 m; non-germline *BRCA* cohort: 9.3 vs. 3.9 m (*p* < 0.001)

EOC, epithelial ovarian cancer; PFS, progression-free survival; b.d., bis die (twice daily); CTH, chemotherapy; m, months.

Despite various treatment options, recurrent EOC is common, occurring in 20–25% of stage I or II patients and in 70% of advanced-disease patients. The first relapse can occur from a few months to five years after treatment and the median recurrence is 18 to 24 months [130]. Histology types do not appear to have a correlation with the recurrence rates [131]. The treatment of recurrent EOC is decided through platinum sensitivity based on the progression-free interval. If there is a response to platinum-based chemotherapy and the progression-free interval is over 6 months, the patients are considered to be platinum-sensitive [130]. Conversely, patients are considered to be platinum-resistant if the progression-free interval is less than 6 months, and refractory if less than 3 months [130,132]. For these patients, non-platinum chemotherapy may be used to increase the platinum-free interval, e.g., single-agent topotecan. This allows a better response to the platinum challenge at a later point [132]. A carboplatin-based combination is generally used for a platinum-sensitive disease; the options include gemcitabine, paclitaxel, and doxorubicin [130]. Surgery may be performed as a secondary cytoreductive surgery and more than 30% of these could involve a bowel resection; each patient should have specific treatment plans depending on their progression-free interval and prognostic factors [133].

## 5. Conclusions

Ovarian cancer is heavily influenced by hereditary factors and up to 25% of patients carry PVs in several genes. These PVs translate the proteins in various DNA repair pathways. The most common mutations are *BRCA1/2* mutations with rarer mutations in other genes such as *BRIP1*, *RAD51C*, *RAD51D*, and MMR proteins. Most of the proteins are implicated in HR repair; treatments to target this pathway include PARP inhibitors, which allow highly specific chemotherapy planning. NGS has a high accuracy for detecting all mutation types as it allows the rapid testing of large genomic regions and identifies lower penetrance genes. Testing can also detect whether the mutations are somatic or germline. NGS is cost-effective as the cost is similar to single-gene sequencing. Testing for mutations allows asymptomatic carriers to be counselled and proactively managed as opposed to regular surveillance, which may not always be successful. More research is needed on the rarer mutations present in ovarian cancer as understanding and detecting the different mutations benefits from genetic counselling and improves the treatment options for ovarian cancer in both the immediate and longer future.

## Figures and Tables

**Figure 1 ijerph-19-08113-f001:**
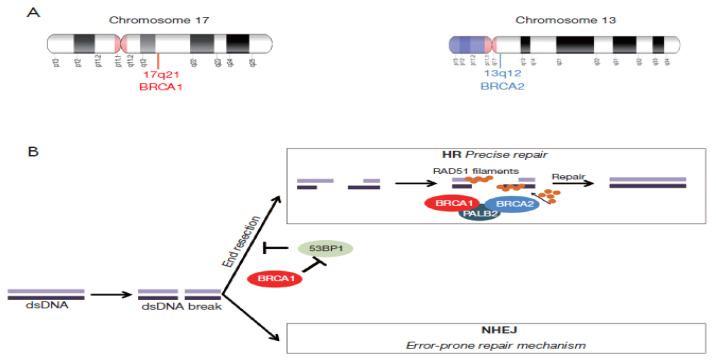
Overview of the roles of *BRCA1* and *BRCA2* in the DNA repair mechanism. (**A**). *BRCA1* and *BRCA2* loci on chromosomes 17 and 13, respectively. (**B**). The initiation of the double-stranded DNA (DSBs) break correction starts with *BRCA1* binding to the site of damage, thus initiating the precise repair via homologous repair (HR) and preventing non-homologous end joining (NHEJ).

**Figure 2 ijerph-19-08113-f002:**
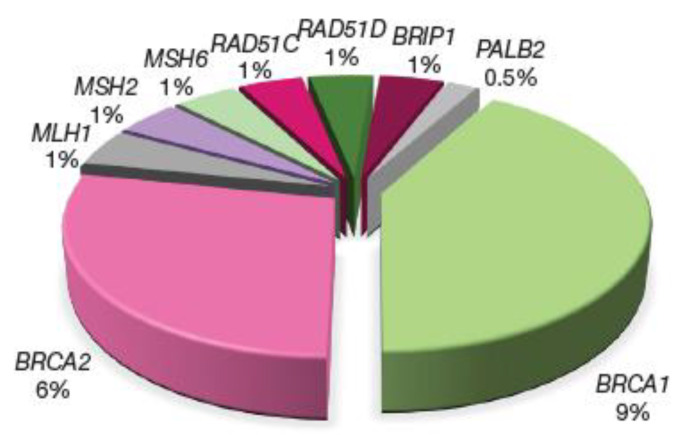
Distribution of germline pathogenic variants (PVs) identified in unselected epithelial ovarian cancer (EOC) patients. Among 21.5% of those PVs, 15% represent alterations to *BRCA1/2* genes and 3.5% of genetic aberrations in other genes compromise the homologous recombination (HR) pathway whilst 3% are PV genes involved in the DNA mismatch repair (MMR) pathway.

**Table 1 ijerph-19-08113-t001:** Common mutations according to ovarian cancer histologies.

Genes Affected (%)	Histology		
	HGSOC	EnOC	CCOC
*BRCA1*	8.0	-	-
*BRCA2*	4.0	5.1	5.1
*ATM*	4.0	9.1	17.9
*BRIP1*	2.0	2.0	-
*PALB2*	-	2.0	2.6
*RAD50*	-	1.0	2.6

HGSOC, high-grade serous ovarian cancer; EnOC, endometrioid ovarian cancer; CCOC, clear cell ovarian cancer.

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
