# Peer review of "Epithelial Ovarian Cancer: Providing Evidence of Predisposition Genes"

_ijerph, 2022, doi:10.3390/ijerph19138113_

Round 1

Reviewer 1 Report

The paper discusses susceptibility to EOC, testing and EOC risk assessment in the context of NGS. The topic is important for better management and treatment of EOC. My comments are listed below:

1. T cell repertoire sequencing and T cell repertoire alteration should be reviewed and discussed for susceptibility to EOC and testing.

2. Molecular traits of genomic alteration and T cell repertoire in the five major EOC histotypes may differ. Please specify mutations that are common among the five histotypes and mutations that are histotype-specific.

3. Please also review recurrent EOC.

4. Please discuss possible treatment options associated with EOC testing outcomes.

Author Response

I am pleased to resubmit for publication the revised version of our manuscript entitled “The puzzle of ovarian cancer genetic counseling”.

We are thankful for the guidance provided by the reviewers to improve the article further and hope you agree with the final revisions. All the reviewers’ comments have been addressed in the revised manuscript and we have provided a point-by-point response to the reviewers’ comments below. All corresponding changes have been marked in blue in the manuscript.

Reviewer 1:

The paper discusses susceptibility to EOC, testing and EOC risk assessment in the context of NGS. The topic is important for better management and treatment of EOC. My comments are listed below:

  1. T cell repertoire sequencing and T cell repertoire alteration should be reviewed and discussed for susceptibility to EOC and testing.

Response:

Thank you for your comments. T cell repertoire sequencing and role of tumour infiltrating lymphocytes have now been discussed.

  1. Molecular traits of genomic alteration and T cell repertoire in the five major EOC histotypes may differ. Please specify mutations that are common among the five histotypes and mutations that are histotype-specific.

Response:

Thank you. Mutations across histologies of ovarian cancer have now been included and also depicted in a table 1. 

  1. Please also review recurrent EOC.

Response:

Thank you. Recurrent EOC has now been discussed.

  1. Please discuss possible treatment options associated with EOC testing outcomes.

Response:

Thank you. PARP inhibitors and risk-reducing options in carriers have now been discussed further.

Reviewer 2 Report

ijerph-1718139  

General comment

This review article is about ovarian cancer and its cancer predisposing gene, but there are relatively very few references to cancer genetic counseling in the title. The manuscript is also redundant, lacking figures and tables, and lacking in novelty compared to papers already published. The manuscript needs to be resubmitted after substantial editing of the content.

Major concerns.

  1. Genes involved in homologous recombination and MMR are listed as cancer predisposition genes, but they are described in parallel and appear to be confused by the authors.For example, as per Line 265, it says VARIOUS MMR PATHWAYS, but the meaning of this phrase is unclear. Also, I think the HR related gene mutation is associated with HGSC, but is there any mention of which tissue type the MMR gene is associated with?
  2. The manuscript cites many of the authors' own review articles, and similar statements are found throughout the manuscript, so it reads as if they were pieced together.
  3. Some of the references cited are probably inappropriate; Ref 40 in Lines 80-82 does not mention PARP inhibitor at all. Whether all references are properly cited is a matter for the authors to check first.
  4. Font sizes are not consistent, making the manuscript appear odd.
  5. No figures or tables, only lengthy text that is labor intensive to read.

Author Response

I am pleased to resubmit for publication the revised version of our manuscript entitled “The puzzle of ovarian cancer genetic counseling”.

We are thankful for the guidance provided by the reviewers to improve the article further and hope you agree with the final revisions. All the reviewers’ comments have been addressed in the revised manuscript and we have provided a point-by-point response to the reviewers’ comments below. All corresponding changes have been marked in blue in the manuscript.

Reviewer 2:

This review article is about ovarian cancer and its cancer predisposing gene, but there are relatively very few references to cancer genetic counseling in the title. The manuscript is also redundant, lacking figures and tables, and lacking in novelty compared to papers already published. The manuscript needs to be resubmitted after substantial editing of the content.

Major concerns.

1. Genes involved in homologous recombination and MMR are listed as cancer predisposition genes, but they are described in parallel and appear to be confused by the authors. For example, as per Line 265, it says VARIOUS MMR PATHWAYS, but the meaning of this phrase is unclear. Also, I think the HR related gene mutation is associated with HGSC, but is there any mention of which tissue type the MMR gene is associated with?

Response

Thank you. This line has been amended. Histologies that the MMR mutations are associated with have been discussed. (section “Molecular landscape”, Lines 131-144). 

2-3. The manuscript cites many of the authors' own review articles, and similar statements are found throughout the manuscript, so it reads as if they were pieced together. Some of the references cited are probably inappropriate; Ref 40 in Lines 80-82 does not mention PARP inhibitor at all. Whether all references are properly cited is a matter for the authors to check first.

Response

Thank you for your comment. We appreciate your thorough evaluation. Indeed, we have worked intensively in the field and cited part of our work. The context of the revised manuscript has been updated and reorganized, based on the reviewers' valuable comments. With this regard, the addition of new references has improved the final list of them.  (Ref. 61-66, 104-111 and 120-133).

4. Font sizes are not consistent, making the manuscript appear odd.

Response

Thank you. The font sizes have now been made consistent.

5. No figures or tables, only lengthy text that is labor intensive to read.

Response

Thank you. Tables and figures have now been added.

Round 2

Reviewer 1 Report

The authors answered most of me questions. It is ready to be published by IJERPH.

Author Response

Dear Editor and Reviewers,

I am pleased to resubmit for publication the revised version of ijerph-1718139 manuscript, entitled “Epithelial Ovarian Cancer – Providing Evidence of Predisposition Genes”.

Thankfully the reviewers provided us with a great deal of guidance, regarding how to better position the article. We are hopeful you agree that this revision will update our comprehensive review. All the comments have been addressed, as shown in the revised version of the manuscript, along with this point-by-point response to the reviewers' comments.

All corresponding are red changes in the second round revision of the manuscript.

Reviewer #1:

General comment:

The authors answered most of me questions. It is ready to be published by IJERPH.”.

Response:

Thank you very much for your positive feedback. We appreciate the opportunity to revise our work for consideration for publication.

Reviewer 2 Report

As in the attached document, there is no response to the comments before the Major concerns and they have not been adequately corrected.

"This review article is about ovarian cancer and its cancer predisposing gene, but there are relatively very few references to cancer genetic counseling in the title. The manuscript is also redundant, lacking figures and tables, and lacking in novelty compared to papers already published. The manuscript needs to be resubmitted after substantial editing of the content".

The discrepancy between the title of the paper and the content is the most significant issue, and the similarity of the content to several review papers the authors have already published is also very concerning.  There is no difference between this paper and the previous ones.

Author Response

Dear Editor and Reviewers,

I am pleased to resubmit for publication the revised version of ijerph-1718139 manuscript, entitled “Epithelial Ovarian Cancer – Providing Evidence of Predisposition Genes”.

Thankfully the reviewers provided us with a great deal of guidance, regarding how to better position the article. We are hopeful you agree that this revision will update our comprehensive review. All the comments have been addressed, as shown in the revised version of the manuscript, along with this point-by-point response to the reviewers' comments.

All corresponding are red changes in the second round revision of the manuscript.

Reviewer #2:

General comment:

As in the attached document, there is no response to the comments before the Major concerns and they have not been adequately corrected.

"This review article is about ovarian cancer and its cancer predisposing gene, but there are relatively very few references to cancer genetic counseling in the title. The manuscript is also redundant, lacking figures and tables, and lacking in novelty compared to papers already published. The manuscript needs to be resubmitted after substantial editing of the content".

The discrepancy between the title of the paper and the content is the most significant issue, and the similarity of the content to several review papers the authors have already published is also very concerning. There is no difference between this paper and the previous ones.”.

Response:

We appreciate you taking the time to offer us your comments and insights related to the paper. Thank you for your constructive feedback. We tried to be responsive to your concerns as we approached our revision. The major concerns have been addressed and there was a point-by-point response in the first round revision.

In terms of the discrepancy between the title of the paper and the content that you kindly raised as the most significant issue, this has now been addressed. The academic editor also recommended us to modify the title of the paper, in order to be clear that this is a review of the predisposing gene in EOC. The revised title of the manuscript is “Epithelial Ovarian Cancer – Providing Evidence of Predisposition Genes”. Moreover, we have added figures and tables in the revised version, whereas for the “lacking in novelty” statement, I have to clarify that the manuscript was an invited review, rather than a research article.

Technically, an abstract had been evaluated a long time before the submission of the manuscript and the conclusion was that it falls within the scope of the special issue “Ovarian Cancer: Prevention and Treatment”. No concern was back then raised related to similarity with already published papers in the literature and based on the abstract the editorial office encouraged the authors to prepare that specific article.

We finally believe that the editor and the reviewers provided us with a great deal of guidance, regarding how to better position the article. Once again, I express our sincere gratitude for your interest to critically review our paper.